# TEACHERACTIVITYNET: A NOVEL DATASET FOR MONITORING FACULTY ACTIVITIES IN OFFICE SETTINGS

## ABSTRACT

In this paper, we introduce a novel dataset for monitoring the activities of faculty members in academic office environments. Advances in computer vision have enabled the automation of workplace monitoring, particularly in educational institutions, where tracking faculty activities presents significant challenges and ethical considerations. Traditional methods of manual supervision are labor-intensive and prone to human error, underscoring the potential of automated video analysis as a more efficient solution. While substantial progress has been made in Human Activity Recognition (HAR) across various domains, research specifically focused on monitoring faculty activities in office settings is limited. Most existing studies concentrate on classroom and student monitoring, revealing a critical gap in faculty surveillance. This paper seeks to address that gap by introducing TeacherActivityNet, a novel video dataset designed to recognize teachers' activities in academic offices, encompassing nine distinct action classes. We tweak the YOLOv8n architecture to propose our model, Teacher Activity Net (YOLOTAN), which is then fine-tuned using our dataset, achieving an average precision of 74.9%, significantly outperforming benchmark models. A comparative analysis of our dataset and methods against existing solutions highlights the potential of TeacherActivityNet to improve automated faculty monitoring systems. The dataset, trained models, and accompanying code are available at https://tinyurl.com/4ub94phh

## 1 INTRODUCTION

The improvement in different computer vision models has opened new frontiers in the automation of various tasks, including the monitoring and surveillance of workplaces. In educational institutions, monitoring teacher activities such as ensuring safety, improving operational efficiency, and evaluating performance can be complex and time-consuming. Moreover, typical manual supervision methods not only demand significant human resources but are also prone to human errors and inconsistencies. To address these challenges, automated faculty monitoring systems using video analysis can be a promising solution.

In recent years, a considerable amount of work has been done in the field of Human Activity Recognition in various contexts such as sports (Host & Ivašić-Kos, 2022)(Xiao et al., 2023), pose estimation (Atikuzzaman et al., 2020), crime scene detection (J & Thinakaran, 2023), healthcare (Gupta et al., 2022), etc. While solving the problem of human activity recognition from videos, the researchers primarily focus on a few aspects - analyzing the applicability of Machine Learning (ML) and Deep Learning (DL) methods in different HAR tasks, creating new datasets for domain-specific HAR-related tasks, and proposing novel solutions for diverse HAR problems. Educational organizations have been the center of experimentation to introduce automation using various AI methods to streamline processes (Ben Williamson & Potter, 2023). Dimitriadou & Lanitis (2023) have critically analyzed the use of AI and emerging technologies in the classroom and recommended that the computer vision-based surveillance system can ensure the safety and security of the classroom alongside tracking the students' participation and attendance. Additionally, a smart surveillance system can help teachers in plagiarism detection and student supervision in an online setting (Saini & Goel, 2019). These systems can also help in identifying abnormal activities on the school premises through CCTV footage analysis (Liu et al., 2023).

However, the intelligent video surveillance systems in educational organizations primarily focus on classroom and student activity monitoring. There has been a lack of existing work on monitoring teachers' activities in an office setting. On the other hand, to solve the HAR problems, the majority of the researchers have adopted machine learning and deep learning-based techniques like CNN, RNN, LSTM, SVM, Naive Bayes, older versions of You Only Look Once (YOLO), and so on. The use of the latest version of YOLO might have good prospects in office surveillance. Our work aims to address this gap in how YOLO would efficiently recognize the teachers' activities in their offices. Our main contributions are:

- Creating a video dataset of teachers' activities in their offices which consists of nine action classes.
- Modifying YOLOv8n to build an efficient detection model YOLOTAN to monitor teachers' activities.

In the next sections, we critically reviewed the state-of-the-art in HAR followed by discussing our datasets and proposed method. Section 4 presents our result followed by the conclusion.

## 2 RELATED WORK

Human Action Recognition (HAR) is a critical research area with applications across various domains such as security or surveillance, sports, education, and more. Many researches were done throughout the years to improve the prediction and recognition task of a variety of activities.

Yuganthini et al. (2021) proposed a wireless method named the Zigbee technique to track employee activity using computer vision. The dataset used in this process consists of videos collected from CCTV cameras in the workplace. In order to find the efficiency of their system, they compared their measured time with the actual in and out time collected from bio-metric entry. However, the system is only supposed to function for a specific region. Another study of the same year, Sikder & Nahid (2021) introduced a dataset, the KU-HAR for heterogeneous HAR. The dataset was created from videos featuring 90 participants performing 18 different actions. Using a Random Forest (RF) classifier, they achieved a precision of 90%.

Later on, many studies were done focusing on the learning or teaching environments of different educational institutions. Rashmi et al. (2021) detected the student actions were performed in computer laboratories. They collected 688 image frames from CCTV cameras installed in the labs and gathered 54,862 samples from these frames for five different action classes. YOLOv3 was used as a method to detect the actions. In the same year Zhao et al. (2021) predicted teacher-student behavior by analyzing various student actions in the classroom during teaching. Some very recent studies also reflected to contribute in the same track. Wang et al. (2024) evaluated teaching quality in real-time by predicting the students' "head-up rate" using YOLOv5. This approach generated more effective results in education quality assessment than traditional survey questionnaires. In order to boost the process and have more reliable outputs, different models are also being proposed in recent studies. Dey et al. (2024) proposed the AdaptSepCX Attention Network model to detect student actions in online education. Their model achieved a high validation accuracy of 92.73%. Moreover, Pabba & Kumar (2024) proposed a vision-based student engagement model, focusing on seven action classes based on students' facial expressions. They used the Multi-task Cascaded Convolutional Networks (MTCNN) method for facial recognition.

Additionally, the activity recognition process was introduced to detect anomalies in fitness activities. Yang et al. (2023) used Pose-Based Branch (PBB) and RGB-Based Branch (RBB) features separately with CNN, ResNet152, and 3D-CNN models to compare. MPOSE-2019, Body Movements-Based Dataset (BMbD), Multi-target Body Movements-Based Dataset (M-BMbD), and the Joint Body Movements and Object Position-Based Dataset (JBMOPbD) were used in this study. The pose-based method outperformed the RGB-based approach. As a final result, the Pose-Based Branch (PBB) outperformed the other one as a feature consideration.

Moreover, some other activities related to security and the prediction of anomalies or suspicious events are introduced in this research area. Singh et al. (2020) proposed CNN and RNN-based models with Inception V3 to predict crime scenes in the UCF crime dataset which contains 1,800 videos. In this study, it was assured that considering a larger dataset and using augmentation played

a vital role in the prediction process. In the same year Shreyas et al. (2020), proposed 3D CNN for anomalous human activity detection using the same dataset. This method outperformed SVM and binary classifiers. Nale et al. (2021) detected suspicious human activity using pose estimation and LSTM on the NTU-D 60 dataset by evaluating the geometrical relations of skeletal joints. Liu et al. (2023) proposed a method for recognizing abnormal behavior on campus using Temporal Segment Transformers (TST), Video Swin Transformer method for the CABR50 dataset. Another research of the same year conducted by Nandhini & Thinakaran (2023) focused on crime scene detection using the stacked hourglass method and Gaussian classifier. This study achieved an accuracy of more than 90% in the test set. Some other methods in the related field named Enhanced Convolutional Neural Network (ECNN) to predict suspicious actions in video surveillance were also proposed as a model (Selvi et al., 2022). Multiple datasets including CCTV footage, CAVIAR, DCSASS, and public datasets were used for this study. Many recent studies were also found related to security issues such as a system developed to predict the possible anomalies in smart homes (Rahman et al., 2024), YOLOv3 to detect real-time suspicious activity in ATM surveillance videos (Menaka et al., 2024) etc. Activation functions such as OP-Tanish activation were also introduced with 1D-CNN with the success of outperforming the basic activation functions such as ReLU and SWISH (Ankalaki & Thippeswamy, 2024).

The application of human activity recognition plays a vital role in sports as well. In recent years, there has been a significant rise in research on sports-based videos. Host & Ivašić-Kos (2022) used some ML and DL-based techniques for activity detection in sports videos. However, it mostly covered the sports played using balls. A custom dataset was prepared combining various existing datasets like THETIS. In the same year, Latha et al. (2022) proposed CNN and LSTM using the UCF-50 video dataset. Xiao et al. (2023) proposed deformable convolution and an adaptive multi-scale feature method. They analyzed sports videos from the UCF Sports, UNF 50, and YouTube (UCF 11) datasets. Goh et al. (2023) focused on fault detection during badminton matches using the YOLOv5 model. The used dataset consisted of 1,900 images from videos, and the model achieved higher accuracy than human judges.

## 3 METHODOLOGY

Our work consists of three major phases - creating the dataset, modifying the YOLOv8 model architecture to obtain better better-performing detection model, and proposing a method for real-time prediction. Figure 1 presents the step-by-step descriptions of our work. Initially, we curate our dataset followed by preprocessing it. After that, we make modifications to the YOLOv8n model architecture to introduce the YOLO Teacher Activity Net (YOLOTAN). We train and fine-tune the YOLOTAN model to predict the action classes from the videos of the faculty members. Besides, we measure action class-wise time for each faculty member through face recognition.

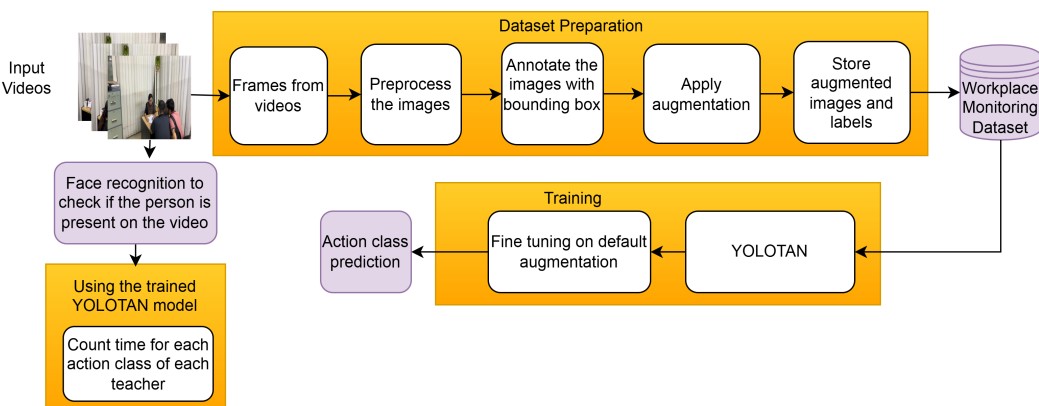

Figure 1: Workflow of our proposed method

## 3.1 DATASET

One of our major contributions is the creation of a dataset that includes a wide range of activities for the task of teacher monitoring. To accomplish this, we collected videos of 19 participants for 9 action classes. The action classes are Arriving, Counselling, Eating, Idle, Leaving, Sleeping, Talking, Using_Phone and Working. The participants were asked to sign an informed consent form before making the videos and an appropriate opt-out policy was followed in every step of the data collection process. As per the General Data Protection Regulation (GDPR) Union (2016), data privacy and security are prioritized. For the diversification of the dataset, the videos were taken from 8 different rooms. The videos were recorded using an iPhone 11 Pro Max, with varying camera distances based on room size and diverse angles capturing multiple participants' actions. The cameraman's average height was 5'6". The dataset includes 19 participants (3 female), mostly aged 20–22, with some around 30. All actions were controlled and guided.

### 3.1.1 COLLECTION

The dataset creation process took place primarily in two stages - training and validation, and testing. In the first stage, we took a sample of 12 from the participants and they were asked to perform one action at a time. Nine videos for each of the action classes per participant were recorded. After collecting the training videos, we took a sample of size 3 from the remaining 7 participants and asked them to do the same. These video sets of three participants across the nine classes are used as the validation set in our experiments.

In the second stage, we made 10 videos of the remaining 4 participants who were not involved in the previous stage of the data collection. As we are interested in measuring how accurate our models are in real-time, unlike the previous stage where different actions were done in different videos, we asked the participants to perform the actions continuously for a specific period. To critically test the model performance in real-time, we made most of the videos in such rooms that were not present in the earlier phase of data collection. In Table 1, the descriptions of the action classes including duration and number of instances after annotation are provided.

### 3.1.2 ANNOTATION

After collecting the dataset, we generate the image frames from the training and validation videos using Roboflow[1] through manual annotation. Three annotators performed manual labeling, with a verifier ensuring the accuracy of all annotations. From each frame, a bounding box is drawn to annotate the object with an action class. Extra caution is exercised while annotating the images that look the same but fall into different action classes. For example, in "Leaving" and "Arriving" action classes, there is a moment in the videos when the participants leaving the seat and taking the seat, look almost identical. Another case is while performing "Idle" action, the participants closed their eyes which potentially conflicts with "Sleeping" action. Taking these ambiguous scenarios into consideration, such frames are discarded from the dataset. After annotating, the numbers of instances in the training, validation, and test sets are 6498, 1337, and 808, respectively.

### 3.1.3 DATASET AUGMENTATION AND PREPARATION

The detailed dataset preparation process is presented in Figure 1. To be compatible with YOLOv8, the images are reshaped to $640 \times 640$ pixels. Two different augmentations are also applied to increase the generalizability of the prediction models. Initially, the images are flipped horizontally. And to give the models the ability to predict from CCTV footage, a noise of 0.3% is added. Both augmentation techniques are employed in accordance with the recommendations made by Singh et al. (2020). In Figure 2 and Figure 3, a sample before augmentation and another one after augmentation are presented. Only the training images are augmented and increased from 6498 to 19494 instances.

After annotation, each image has two files saved in two directories - one is an image and another is the label file that contains bounding box coordinates. We use the YOLOv8-oriented bounding box dataset. In this dataset, the label file has 9 values instead of 5 which is common in rectangular

---

[1] https://roboflow.com

Table 1: Action class definition with duration and total instances in training dataset

| Class Name | Class ID | Definition of the Action Class | Duration Per Video | Total Instances |
|---|---|---|---|---|
| Arriving | 0 | Person arriving in the room and taking his/her seat. | 30/60 seconds | 665 |
| Counselling | 1 | Teacher giving counselling time to multiple students. | 30/60 seconds | 728 |
| Eating | 2 | Person eating or drinking. | 10 seconds | 739 |
| Idle | 3 | Idle for some time, doing none of the other mentioned actions. | 30/60 seconds | 755 |
| Leaving | 4 | Person leaving his/her seat and going out of the room. | 5 seconds | 654 |
| Sleeping | 5 | Person sleeping in two positions: laying his/her head on the chair or putting their head down on the table. | 10 seconds | 739 |
| Talking | 6 | Talking via mobile phone (not considering talking to persons). | 30/60 seconds | 727 |
| Using_Phone | 7 | Using his/her phone while sitting in the chair or standing. | 30/60 seconds | 748 |
| Working | 8 | Focused on the computer/laptop while using mouse or keyboard. | 30/60 seconds | 743 |

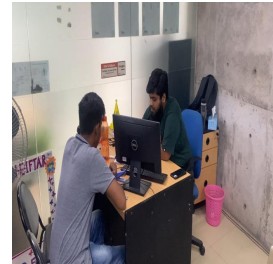 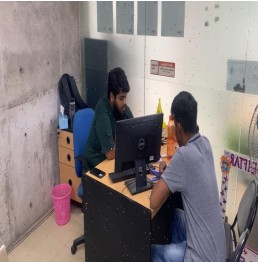

Figure 2: A sample image before augmentation      Figure 3: A sample image after augmentation

bounding box coordinates. The label of our images has quadrilateral coordinates. The first value is the class ID and the remaining values are coordinates of four corners like $(x_1, y_1), (x_2, y_2), (x_3, y_3)$, and $(x_4, y_4)$.

## 3.2 YOLO TEACHER ACTIVITY NET (YOLOTAN)

YOLOv8 is considered state-of-the-art for object detection in real-time. This version of YOLOv8 has more speed and accuracy than the previous versions. A few key components of YOLOv8 are anchor-free detection, multi-scale predictions, and decoupled head architecture. We adopt the YOLOv8n as the base model for our detection task. In YOLOv8n there are three main parts in the architecture - Backbone, Neck, and Head.

In our proposed model, we modify the backbone of the YOLOv8n architecture. In the backbone, there are five feature pyramids (P) and four stages with each having a c2f (Cross-Stage Partial Network with 2 Convolutions and Fusion) module and a convolution module. We modify the last layer of the convolution module with a residual connection. The input of the convolution module of the last layer adds to the output of the same layer. This process is done for every convolution module. The modified convolution module is shown in Figure 4. In Figure4, the internal mechanism of the modified convolution module with residual connection is presented. It adds a residual connection when the input and output channels of the convolution layer are the same with a stride of (1,1). This can help mitigate the vanishing gradient problem and potentially improve the learning of deeper features. Residual connections offer a direct pathway for gradients to propagate backward, helping

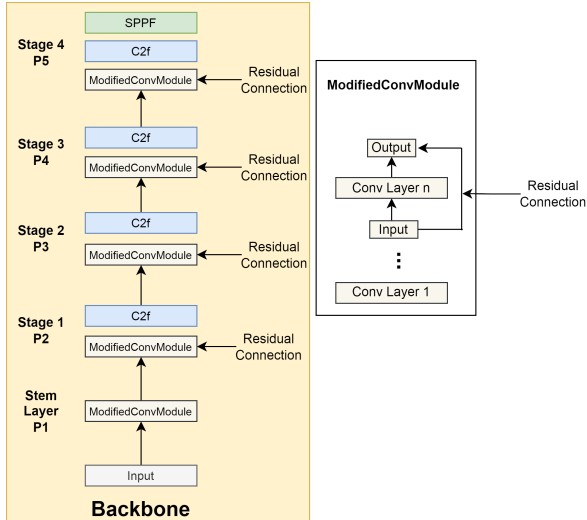

Figure 4: Modified backbone of our proposed YOLOTAN

to alleviate the vanishing gradient problem. Mathematically, considering the gradient of the loss $L$ with respect to the input $x$, we have:

$$\frac{\partial L}{\partial x} = \frac{\partial L}{\partial y} \cdot \left( \frac{\partial F(x)}{\partial x} + 1 \right) \tag{1}$$

The addition of the $+1$ term ensures that the gradients can flow back directly, even when $\frac{\partial F(x)}{\partial x}$ becomes very small. The added residual connections might introduce a small computational overhead, but it is likely to be minimal given that they are only added under specific conditions.

For benchmark analysis, in addition to creating detection models with our proposed YOLOTAN, we use pre-trained and fine-tuned YOLOv8 and fasterrcnn_resnet50_fpn model. All the models are trained on annotated images and labels with bounding box coordinates.

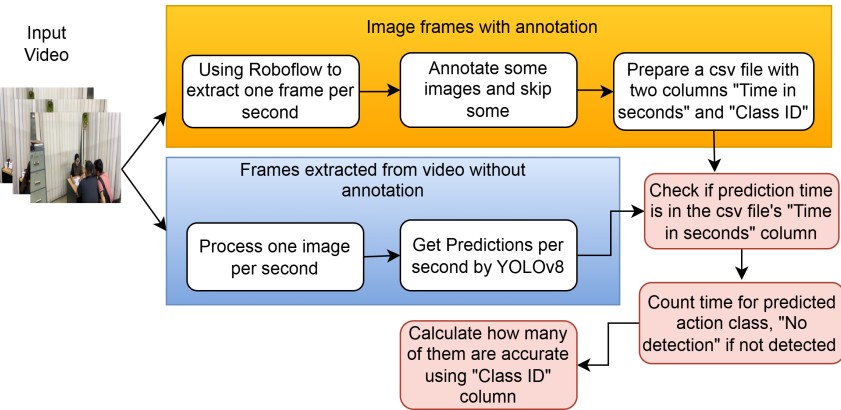

Figure 5: Flow chart for prediction on sample test dataset video with time count for each action class

### 3.3 PREDICTION FROM VIDEOS

A big challenge for any computer vision model is to measure the performance in the real world. In addition to finding the validation and test performance of our trained YOLOTAN model, we measure its detection performance from videos. In Figure 5, the steps to measure this detection performance from test videos are described.

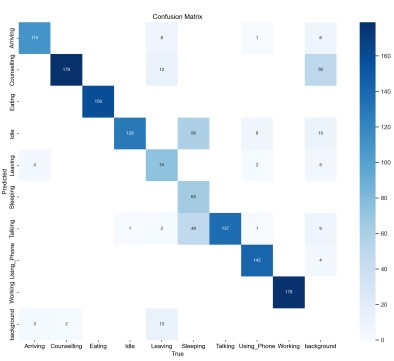

(a) Confusion Matrix for Validation Dataset

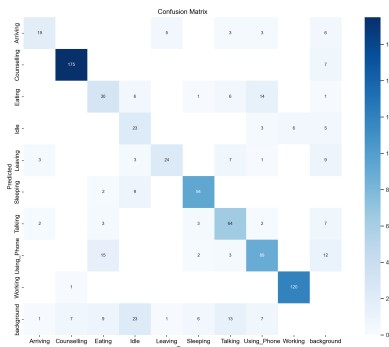

(b) Confusion Matrix for Test Dataset

Figure 6: Confusion Matrix for Validation and Test Datasets

## 4 RESULTS

In this section, we discuss our experimental results. We use a machine with Intel(R) Core(TM) i7-7700 CPU @ 3.60GHz processor with NVIDIA TITAN xp 12GB GPU to train our YOLOTAN models as well as the basic YOLOv8 and faster RCNN models.

### 4.1 TEST AND VALIDATION RESULTS

As we use YOLOTAN as a prediction model for our study, it has two different sets, one is used for training and the other for validation. This method takes images labeled with a bounding box. Our proposed model achieved 0.941 as the mean Average Precision(mAP50) on the validation dataset and 0.749 on the test dataset.

Table 2: Box precision, recall, mAP50 and mAP50-95 results for test dataset using YOLOTAN for each action class

| Action | Images | Instances | Box (Precision) | Recall | mAP50 | mAP50-95 |
|---|---|---|---|---|---|---|
| Arriving | 25 | 25 | 0.539 | 0.760 | 0.615 | 0.382 |
| Counselling | 183 | 183 | 0.967 | 0.948 | 0.988 | 0.636 |
| Eating | 59 | 59 | 0.577 | 0.475 | 0.496 | 0.332 |
| Idle | 64 | 64 | 0.469 | 0.266 | 0.387 | 0.218 |
| Leaving | 30 | 30 | 0.570 | 0.733 | 0.709 | 0.461 |
| Sleeping | 106 | 106 | 0.895 | 0.805 | 0.924 | 0.535 |
| Talking | 96 | 96 | 0.845 | 0.698 | 0.833 | 0.400 |
| Using_Phone | 119 | 119 | 0.781 | 0.782 | 0.813 | 0.396 |
| Working | 126 | 126 | 0.975 | 0.947 | 0.978 | 0.637 |

In Table 2, the achieved results using YOLOTAN for different action classes are shown separately. Box precision, recall, mAP50 and mAP50-95 value for each class is shown in this Table2. The data instances are considered as images with bounding box to generate these test results. From this Table 2, we can observe that the Idle action class has the lowest performance in mAP50 whereas the prediction of Counselling and Working is showing higher prediction performance. Though the used dataset contains fewer instances of Leaving and Arriving action classes, the prediction worked well for these action classes too. The overall mAP50 for the dataset using YOLOTAN is around 0.749 on the test dataset considering all the action classes.

In Figure 6a, the confusion matrix using the YOLOTAN pre-trained model for the validation dataset is provided. Predictions for most of the action classes are very good on the validation dataset. The prediction of the action class "Idle" is average. The main reason behind average accuracy is due to some image frames of "Idle" action class where participants were looking down. It looks like their eyes are closed. So, the model predicts "Sleeping". As we can see 59 image frames predicted as "Sleeping" were actually "Idle".

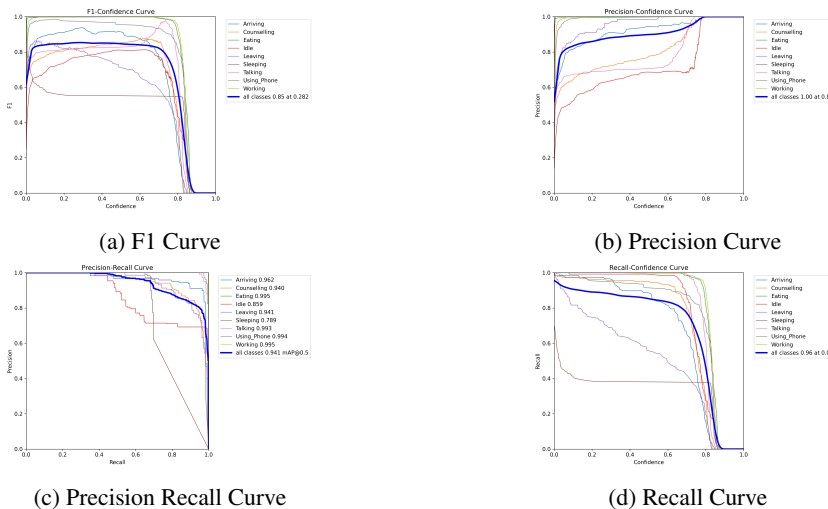

(a) F1 Curve

(b) Precision Curve

(c) Precision Recall Curve

(d) Recall Curve

Figure 7: F1, Precision, and Recall Curves for All Activities

In Figure 6b, prediction accuracy drops in the test dataset. Counselling class has very high accuracy. The Eating class has the lowest accuracy. The main reason behind this is actually the Using_Phone action class. In some of the image frames, we can see the participant holding the food like they are holding the mobile phone. That is why in 14 instances the model is predicting Using_Phone instead of Eating.

The confidence curve for F1 score, precision, precision-recall, and recall is shown in Figure **??**. From Figure 7a, we can find that the confidence threshold for the F1 score for all the classes is 0.282 where the F1 value is 0.85. Figure 7b reflects the average precision value for all classes which is 1.00 at a confidence threshold of 0.806. Again, Figure 7c shows the precision-recall curve reflecting the mean Average Precision(mAP0.5) value in 0.934. Finally, Figure 7d shows a recall curve with an average recall value of 0.96. However, it is observed that for all other values except precision, the average performance of sleeping is much lower than the other action classes.

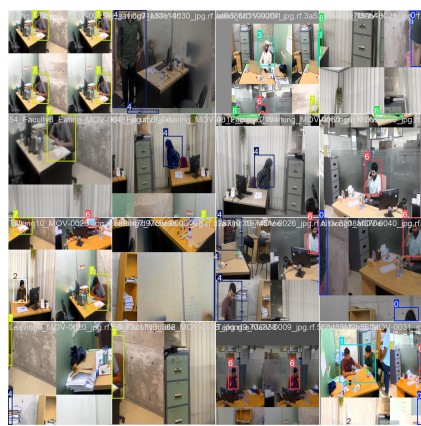
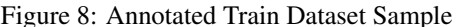

Figure 8: Annotated Train Dataset Sample

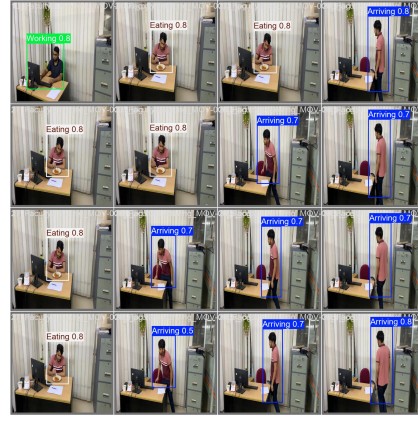

Figure 9: Predictions of Validation Dataset

In Figure 8, training images for the YOLOTAN and Faster RCNN-based pretrained model are provided. As the training images are annotated before use, every image contains an anchor or bounding box. In Figure 9, the predicted images of validation by the YOLOv8 models are shown. The bounding box and the class score are also provided for each activity. The class score reflects how reliable the prediction result YOLOTAN is for each class.

In order to evaluate some other method results using our dataset, we use the Faster RCNN ResNet50 FPN pre-trained model. In Table 3, the results achieved for the Faster RCNN ResNet50 FPN model

Table 3: Accuracy for test dataset using Faster RCNN ResNet50 FPN for each action class

| Action Class | Instances | Correct Prediction | mAP50 |
|---|---|---|---|
| Arriving | 25 | 21 | 0.6602 |
| Counselling | 183 | 170 | 0.9893 |
| Eating | 59 | 30 | 0.1949 |
| Idle | 64 | 3 | 0.2134 |
| Leaving | 30 | 22 | 0.3934 |
| Sleeping | 106 | 68 | 0.7727 |
| Talking | 96 | 58 | 0.8234 |
| Using_Phone | 119 | 39 | 0.6139 |
| Working | 126 | 39 | 0.587 |

are shown in terms of correct predicted instances and thus mAP50. From the table, we can find that the prediction of eating class is around 0.1949 for this model. However, the prediction rate of the Idle class is significantly low for this model in our dataset. Counselling class can be considered as having the highest performance in terms of mAP50. The overall achievement rate of mAP50 for Faster RCNN ResNet50 is 0.58 which is lower than the overall rate of YOLOTAN.

Table 4: Predicted frame counts for real time video as test dataset using the model

| Action Class | Time in Seconds | Annotated Frames | Accurately Predicted Frames |
|---|---|---|---|
| Arriving | 3 | 3 | 1 |
| Counselling | 38 | 29 | 29 |
| Eating | 0 | 7 | 0 |
| Idle | 26 | 4 | 4 |
| Leaving | 8 | 3 | 3 |
| Sleeping | 12 | 12 | 11 |
| Talking | 11 | 8 | 5 |
| Using_Phone | 14 | 10 | 10 |
| Working | 2 | 12 | 2 |
| No Detection | 15 | N/A | N/A |

## 4.2 MODEL PERFORMANCE ON VIDEOS

All the results discussed above are generated using image instances with a specific bounding box. In order to consider the real-time scenario, we take a real video with no boundaries to evaluate the performance of the model for all 9 action classes. Therefore, we take a video of 127 seconds from the test dataset to perform the task. In the first phase of the task, we perform the operation of face recognition to detect the person properly. Employee pictures are used here to operate the recognition task. After the face recognition process is performed successfully, we use our proposed model YOLOTAN to predict the action class for each task. To compare the predicted results and generate the accurately predicted tasks, the video was divided into image frames and annotated using Roboflow. After that, the predicted and annotated image frames are compared to find the correctly predicted instances. In Table 4, the annotated frame numbers and predicted frame numbers are given for each action class. The total predicted time for each action class is also provided to find a decision on overall task prediction for each person. The "No Detection" class indicates the number of times the person was not involved in any defined action class from the available 9 classes.

Table 5: Comparative baseline analysis with our YOLOTAN model

| Metric | YOLOv5 | DSC (YOLOv8n) | SimpleGS (YOLOv8n) | SC (YOLOv8n) | YOLOTAN |
|---|---|---|---|---|---|
| mAP50 | 0.697 | 0.473 | 0.503 | 0.665 | 0.749 |
| Inference Speed | 4.5 ms/image | 2.16 ms/image | 1.82 ms/image | 9.34 ms/image | 2.8 ms/image |

## 4.3 ABLATION STUDIES

In Table 5, a comparison of different convolutional models in YOLOv8 with YOLOv5 and YOLOTAN is shown.

**YOLOv5**: Using YOLOv5 pre trained model, we achieved mAP50 of 0.697 for test dataset. However, the inference speed for processing images is not good.

**Depthwise Separable Convolutional Model(DSC of YOLOv8)**: This method uses DWConv (Depthwise mAP50 is lower than the other two models.

**Simple Grouped Shuffle Convolutional Model(Simple GS of YOLOv8)**: This is a combination of both the Standard Convolutional Model and Depthwise Separable Convolution models for group sampling. We can clearly see that it has the best inference speed 1.82 ms/image compared to the other two methods.

**Standard Convolutional Model(SC Model of YOLOv8)**: This typically refers to a deep learning model using standard convolutional layers for feature extraction. From Table 5, it is observed that this model has outperformed the other two applied methods(DSC and Simgple GS) in evaluation using mAP50 on test dataset. With an mAP50 value 0.749.

**YOLOTAN**: From the Table 5, we can say that our proposed model YOLOTAN demonstrates superior performance than other models with mAP50 of 0.749 on the test dataset. However, the inference speed of the SC model is marginally better than YOLOTAN. Although the YOLOTAN inference speed is average, we can trade a little bit of speed for more accuracy.

## 4.4 ACTIVITY RECOGNITION MODEL PERFORMANCE

Our main task is to recognize the activities of teachers. To achieve this, we have explored several activity recognition models to gain insights into spatial and temporal-based approaches. We analyzed the performance of three models: two of them, MC3-18 and R2plus1D, are based on the ResNet-18 architecture, while the third, the Temporal Segment Transformer (TST), is a custom 3D CNN model built on ResNet3D-18. The TST is enhanced with temporal pooling and a fully connected classifier, designed for efficient spatiotemporal feature learning and action recognition in video data.

Table 6: Comparing activity recognition models

| Metric | MC3-18 | R2plus1D | Temporal Segment Transformer |
|---|---|---|---|
| Validation Accuracy | 48.15% | 40.74% | 70.42% |
| Test Accuracy | 37.11% | 32.85% | 58.34% |

In Table 6, we can see that the performance of the activity recognition models is not satisfactory, with the test dataset performance being notably poor. These models were trained directly on videos. Based on the results, we can conclude that YOLOTAN and FasterRCNN-based pretrained models outperform the activity recognition models.

## 5 CONCLUSION

We present TeacherActivityNet, a novel dataset for monitoring teachers' activities, consisting of videos meticulously recorded in academic office environments. Our proposed model YOLOTAN demonstrates a substantial improvement in average precision over the base model. One of the major limitations of our dataset is that the videos were captured using smartphones to simulate CCTV recordings. Utilizing actual CCTV footage in future work could potentially enhance the model's precision during fine-tuning.

We anticipate that the release of this dataset, along with accompanying resources, will facilitate advancements in human activity recognition, encouraging the development of new datasets and solutions for various computer vision applications.

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
