# OpenReview forum: "TeacherActivityNet: A Novel Dataset for Monitoring Faculty Activities in Office Settings"
_ICLR.cc/2025/Conference — ICLR 2025 Conference Withdrawn Submission_

### Official Review · Reviewer_bQRG · 2024-10-24

**Soundness:** 2
**Presentation:** 1
**Contribution:** 2
**Rating:** 3
**Confidence:** 5

**Summary:**

This paper presents a data set of teacher activity monitoring to solve the problem of lack of teacher monitoring data. The work of this paper has certain significance, but there are big problems, especially in the writing, so it is not accepted for the time being.

**Strengths:**

see summary

**Weaknesses:**

see questions

**Questions:**

1. The innovation points and problems solved in the abstract are not obvious. The author should first introduce the background content, that is, the monitoring of teachers' activities, and then explain the existing problems and the innovation points proposed in this paper. In addition, it can be seen from the abstract that the main work of this paper is to propose the data set and improve YOLOv8n, the latter proposal does not solve the corresponding problems. The length of the abstract is too long, and it is suggested that the author shorten it to less than 400 words.
2. The problem to be solved in this paper is "How to solve YOLO to effectively identify teachers' activities in the office", which is not universal. Given that the title of this article is dataset, the author should verify the reliability of the dataset through multiple methods, not just improve yolo and fine-tune yolo with that dataset.
3. It is suggested that the author expand relevant work and do not only introduce HAR. At the same time, there are too many contents about HAR, and some similar contents are repeated in the introduction.
4. The content of the introduction is weak, and there is no introduction to the problems and methods solved in this paper, especially the part about how to improve yolo.
5, it is recommended that the author check the picture and table format, the table should choose three lines as far as possible, and there should be no empty lines between the table content and the table head, as shown in Table 2 and Table 3. At the same time, adjust the image format. As shown in Figure 7, the subscripts of (a), (b), (c) and (d) are not aligned with the picture, the picture content is fuzzy, and the font is too small to see clearly. It is also suggested that the author beautify the flow chart in Figure 1.
6. I hope the author can add the introduction on how to divide teacher behavior, instead of directly dividing it into 9 actions.
7. The number of data sets provided in this paper is too small, with only 20-120 samples for each action, which lacks reference value.

---

> ### Author Response · Authors · 2024-11-27
> **Response to reviewer bQRG**
>
> #P1: The innovation points and problems solved in the abstract are not obvious. The author should first introduce the background content, that is, the monitoring of teachers' activities, and then explain the existing problems and the innovation points proposed in this paper. In addition, it can be seen from the abstract that the main work of this paper is to propose the data set and improve YOLOv8n, the latter proposal does not solve the corresponding problems. The length of the abstract is too long, and it is suggested that the author shorten it to less than 400 words.
>
> Response: Our abstract is already concise, with a total of only 201 words, well below the 400-words.
>
> #P2: The problem to be solved in this paper is "How to solve YOLO to effectively identify teachers' activities in the office", which is not universal. Given that the title of this article is dataset, the author should verify the reliability of the dataset through multiple methods, not just improve yolo and fine-tune yolo with that dataset.
>
> Response: We tested our dataset using YOLO and Faster R-CNN to evaluate performance. Additionally, we compared it with several activity recognition models, including ResNet-18-based pretrained models (MC3-18 and R(2+1)D) and the Temporal Segment Transformer model.
>
> #P3: It is suggested that the author expand relevant work and do not only introduce HAR. At the same time, there are too many contents about HAR, and some similar contents are repeated in the introduction.
>
> Response: Thank you for your valuable comment. We will try to reduce repeated contents.
>
> #P4: The content of the introduction is weak, and there is no introduction to the problems and methods solved in this paper, especially the part about how to improve yolo. 5, it is recommended that the author check the picture and table format, the table should choose three lines as far as possible, and there should be no empty lines between the table content and the table head, as shown in Table 2 and Table 3. At the same time, adjust the image format. As shown in Figure 7, the subscripts of (a), (b), (c) and (d) are not aligned with the picture, the picture content is fuzzy, and the font is too small to see clearly. It is also suggested that the author beautify the flow chart in Figure 1
>
> Response: Introduction has been updated. And we have followed the table format given in the ICLR template available on overleaf. And figure subscripts are organized by the same template format of ICLR.
>
> #P5: I hope the author can add the introduction on how to divide teacher behavior, instead of directly dividing it into 9 actions.
>
> Response: We believe our 9 action class is more suitable for our work rather than focusing on the teachers behavior with another students or person.
>
> #P6: The number of data sets provided in this paper is too small, with only 20-120 samples for each action, which lacks reference value.
>
> Response: In our dataset we have 6499 samples of only training dataset. We have shown this in Table 1 in our paper.
>
> Note: Updated paper will be uploaded in next 12-15 hours. We are still working on it.

---

### Official Review · Reviewer_7tqZ · 2024-11-01

**Soundness:** 2
**Presentation:** 2
**Contribution:** 3
**Rating:** 5
**Confidence:** 4

**Summary:**

The author mentioned that the existing research had predominantly focused on classrooms and student activities, with a little intention to faulty monitoring or office settings. This effectively sets up a motivation for conducting their research. The paper’s contributions are clearly stated for a creation of a video dataset specifically capturing office activities across 9 action classes and a development of YOLOTAN model.

**Strengths:**

The paper introduces TeacherActivityNet, a new dataset focused on monitoring teacher activities in academic environments. It centered on educational settings which is unique and could open opportunities for specialized applications in performance analysis in office and classroom observation. The proposed YOLOTAN model showcases an effective approach to improving accuracy and speed in the YOLOv8 framework. Tables 3 through 5 present an organized summary of results to easily interpret the performance of various models in different action classes.

**Weaknesses:**

The paper lacks useful information or statistics about camera settings used for recording the videos for the dataset. In the data collection section, the author should describe a detailed of camera setting such as type of camera, field of view, frame rate, resolution, where the camera is set up, the distance between the actors and the camera. Please illustrate the camera setting in the experimental environment. This detail can help readers understand the conditions under which the data was collected and may influence the model performance.

The paper currently does not provide information regarding the age of the actors or the gender distribution (number of females and males) involved in the dataset. Including demographic details such as age and gender is required for understanding the context of the monitored activities.

The current work does not specify whether the actors are asked to perform actions freely or under controlled instructions. For a dataset creation task, it would be good to include whether the actions were performed spontaneously, or guided specific instructions could impact the interpretation of the dataset and subsequence model training and evaluation.

**Questions:**

Please include a more thorough comparison with similar datasets, such as those used for activity recognition in educational or surveillance contexts, to emphasize proposed dataset’s uniqueness or improvement over existing resources.

If possible, please describe more details on the annotation process. It would be beneficial to know if the annotations were verified by multiple annotators or checked for consistency since the annotation quality is important for supervised learning, and inconsistencies could impact model accuracy.

Although YOLOTAN is compared to other YOLO models, comparisons to non-YOLO models commonly used in action recognition, such as those based on CNN-RNN architectures or transformer-based models, would provide a better assessment of YOLOTAN’s relative strengths and weaknesses.

**Details Of Ethics Concerns:**

The paper does not discuss the ethical or privacy implications of recording activities where privacy may be a concern. A brief discussion on data privacy and consent would be valuable to address ethical concerns.

---

> ### Author Response · Authors · 2024-11-27
> **Response to reviewer 7tqZ**
>
> P1#: The paper lacks useful information or statistics about camera settings used for recording the videos for the dataset. In the data collection section, the author should describe a detailed of camera setting such as type of camera, field of view, frame rate, resolution, where the camera is set up, the distance between the actors and the camera. Please illustrate the camera setting in the experimental environment. This detail can help readers understand the conditions under which the data was collected and may influence the model performance.
>
> Resposne: All the videos were recorded using an iPhone 11 Pro Max, with the distance from the camera varying based on the room size. To ensure diversity, we captured different angles while recording multiple participants' actions in the same room. The average height of the cameraman was approximately 5'6".
>
> P2#: The paper currently does not provide information regarding the age of the actors or the gender distribution (number of females and males) involved in the dataset. Including demographic details such as age and gender is required for understanding the context of the monitored activities.
>
> Response: Most participants in the dataset are between 20 and 22 years old, with a few around 30 years of age. Out of 19 participants, 3 were female.
>
> P3#: The current work does not specify whether the actors are asked to perform actions freely or under controlled instructions. For a dataset creation task, it would be good to include whether the actions were performed spontaneously, or guided specific instructions could impact the interpretation of the dataset and subsequence model training and evaluation.
>
> Response: All the actions performed by the participants were controlled and guided.
>
> P4#: Please include a more thorough comparison with similar datasets, such as those used for activity recognition in educational or surveillance contexts, to emphasize proposed dataset’s uniqueness or improvement over existing resources.
>
> Response: We thank the reviewer for highlighting this crucial point. In the next phase of our work, we will aim to address similar dataset comparison tasks.
>
> P5#: If possible, please describe more details on the annotation process. It would be beneficial to know if the annotations were verified by multiple annotators or checked for consistency since the annotation quality is important for supervised learning, and inconsistencies could impact model accuracy
>
> Response: We employed three manual annotators, with one additional person responsible for verifying all the annotations.
>
> P6#: Although YOLOTAN is compared to other YOLO models, comparisons to non-YOLO models commonly used in action recognition, such as those based on CNN-RNN architectures or transformer-based models, would provide a better assessment of YOLOTAN’s relative strengths and weaknesses.
>
>
> Response: We compared YOLOTAN with the Faster R-CNN model and, more recently, with ResNet-18-based pretrained models MC3-18 and R(2+1)D, as well as the Temporal Segment Transformer model.
>
> Note: Updated paper will be uploaded in next 12-15 hours. We are still working on it.

---

> > ### Comment · Reviewer_7tqZ · 2024-12-03
> > **Thanks to Authors**
> >
> > Thank you very much for your response and the effort put into addressing the comments. I appreciate the updates made to the paper. However, the experiments presented still fall short in fully demonstrating the effectiveness of the proposed dataset. Although the improvements are evident, the experimental validation could benefit from additional evidence to strengthen the case for the dataset’s utility. Therefore, I would like to maintain my rating of your work.
> >
> > Additionally, I would like to raise the following points:
> > While the authors have provided more details on the camera used, I still find that crucial information such as the frame rate and resolution has not been explicitly stated in the dataset.
> >
> > Although the authors clarified that the actions were guided and controlled, I still have concerns regarding the representativeness of the dataset. If the actions were scripted or heavily guided, this could limit the ability of the dataset to reflect real-world behavior, where actions tend to be more spontaneous and varied. The authors do not provide sufficient detail on how exactly participants were instructed to act, which is crucial for evaluating the applicability of the dataset to real-world scenarios.
> >
> > While the authors mention using multiple annotators and a verification process, I still suggest a more thorough evaluation of annotation quality for supervised learning task.

---

> > > ### Author Response · Authors · 2024-12-03
> > > **Reposne to reviewer 7tqZ(2)**
> > >
> > > Thank you for your valuable feedback and for taking the time to review our work in detail. We sincerely appreciate your constructive comments and your recognition of the improvements made to the paper.
> > >
> > > Regarding the frame rate and resolution, we would like to clarify that our dataset comprises videos of varying lengths across different action classes. Consequently, the frame rates range from 1 fps to 12 fps to accommodate this variability. The camera resolution was consistently set to HD at 60 fps for capturing the dataset.
> > >
> > > In relation to your concerns about the guided or scripted nature of the actions, we would like to emphasize that participants were only instructed on which actions to perform, but the manner in which they performed these actions was left to their discretion. This approach allowed for considerable diversity, as highlighted in the paper. We believe that this diversity reflects real-world behaviors to a large extent. Furthermore, in our test dataset, actions were recorded in a spontaneous manner to add realism. Additionally, some participants were faculty members from a reputable private university, bringing their expertise and familiarity with realistic behavior into the recordings.
> > >
> > > Finally, while we utilized multiple annotators and implemented a verification process to ensure annotation quality, we acknowledge the importance of further evaluation. In the next phase of our work, we plan to incorporate additional annotators and verifiers to strengthen the robustness and reliability of the dataset.
> > >
> > > Once again, we thank you for your insightful comments and for helping us improve the quality of our work. We remain committed to addressing these points in our ongoing and future efforts.

---

### Official Review · Reviewer_baDe · 2024-11-03

**Soundness:** 1
**Presentation:** 1
**Contribution:** 1
**Rating:** 1
**Confidence:** 5

**Summary:**

The paper introduces a video dataset to monitor faculty activities in an academic office setting. It captures nine action classes e.g., Arriving, Counselling, Idle, Working, using a cell phone camera, recorded from 19 participants in office spaces. To recognize these activities, the authors propose YOLOTAN - a modified YOLOv8n model architecture with added residual connections to improve performance in activity recognition.

**Strengths:**

The paper proposes to explore activity recognition in a novel office-like setting.

**Weaknesses:**

Ethical and privacy concerns have not been addressed - The paper does not address the underlying ethical and privacy implications of faculty monitoring. Camera surveillance and automatic activity recognition in office settings raises significant concerns particularly in an academic environment. Even if we assume that the dataset was collected with permission, this should be explicitly mentioned in the manuscript.

Little or incremental technical contribution:
- Slight modification to YOLO architecture by adding residual connections. No ablation study to explain why this would work for activity recognition.
- Lack of temporal modeling for video based activities.
- Lack of significant gains over comparative models.
- Limited and non-diverse dataset.

**Questions:**

1. What is the accuracy of state of the art video activity recognition models on the proposed dataset.
2. How well does the proposed YOLOTAN method generalize to other video-based datasets on the problem of activity recognition.

**Details Of Ethics Concerns:**

The paper does not address the underlying ethical and privacy implications of faculty monitoring. Camera surveillance and automatic activity recognition in office settings raises significant concerns particularly in an academic environment. Even if we assume that the dataset was collected with permission, this should be explicitly mentioned in the manuscript.

---

> ### Author Response · Authors · 2024-11-27
> **Response for reviewer baDe**
>
> P1#: Ethical and privacy concerns have not been addressed - The paper does not address the underlying ethical and privacy implications of faculty monitoring. Camera surveillance and automatic activity recognition in office settings raises significant concerns particularly in an academic environment. Even if we assume that the dataset was collected with permission, this should be explicitly mentioned in the manuscript.
>
>  Response: First of all, I would like to thank the reviewer for their valuable feedback. The ethical concern raised by the reviewer has been addressed and is explicitly mentioned in the paper.
> To ensure data privacy in this research, we have implemented the following measures:
> Consent was obtained from every individual involved in the data collection process.
> Adherence to data privacy and respect for individual rights to make informed choices has been ensured.
> We strictly follow the General Data Protection Regulation (GDPR) guidelines to protect personal data and uphold individual privacy when working with real data.
>
> P2#: Slight modification to YOLO architecture by adding residual connections. No ablation study to explain why this would work for activity recognition.
>
> Response:
> We have presented the accuracy with and without the residual connection in Section 4.3 and Table 5. SC (YOLOv8n) refers to the model without the modification in the YOLO backbone, while YOLOTAN represents the model with the modification.
>
>
> P3#: Lack of temporal modeling for video based activities / What is the accuracy of state of the art video activity recognition models on the proposed dataset.
>
>
> Response: We experimented with the ResNet-18-based pretrained models MC3-18 and R(2+1)D, as well as the Temporal Segment Transformer model for temporal modeling. However, their accuracy was significantly lower compared to YOLO or Faster R-CNN.
>
> P4#: Lack of significant gains over comparative models.
>
> Response: In our study, we evaluated the performance of YOLOv8n and our proposed model, YOLOTAN, using the mean Average Precision at an Intersection over Union (IoU) threshold of 0.50 (mAP50). YOLOv8n achieved an mAP50 of 66.5%, while YOLOTAN demonstrated a significant improvement with an mAP50 of 74.9%. This enhancement underscores the effectiveness of the modifications implemented in YOLOTAN.
>
> P5#: Limited and non-diverse dataset.
>
> Response: In Section 3.1, we mentioned that our dataset includes multiple rooms to ensure diversity. Additionally, some action classes encompass various action types to further enhance the dataset's diversity.
> For example, the "Talking" action class includes talking on the phone with multiple styles, such as standing, sitting, or holding the phone close to the ear or mouth. The "Sleeping" action class incorporates two styles: leaning the head back on a chair and resting the head on a desk. Similarly, the "Eating" action class covers several eating styles.
> Overall, all action classes include diverse action styles to represent a wide range of variations. We have tried different angles when recording multiple participants' actions in the same room as well.
> We agree with the reviewer’s comment, however, our dataset has 19494 image frames after augmentation which is substantial for our task.
>
> P6#: How well does the proposed YOLOTAN method generalize to other video-based datasets on the problem of activity recognition
>
> Response: We appreciate the reviewer highlighting the crucial point of comparing YOLOTAN’s performance on other publicly available video datasets. We will make efforts to address this in the next phase of our work.
>
> Note: Updated paper will be uploaded in next 12-15 hours. We are still working on it.

---

### Note · Authors · 2025-01-23

I have read and agree with the venue's withdrawal policy on behalf of myself and my co-authors.